# Breastfeeding complexities and sociocultural barriers in the context of preventing perinatal transmission of HIV: A descriptive phenomenology in Northern Ghana

**Awinaba Amoah Adongo**[ID]*, **Kofi Osei Akuoko, Jonathan Mensah Dapaah, Esmeranda Manful**

Department of Sociology and Social Work, Kwame Nkrumah University of Science and Technology, Kumasi, Ghana

* awinabaadongo@gmail.com

## Abstract

### Purpose

Although World Health Organization (WHO) policies aim to promote exclusive breast-feeding and replacement feeding for HIV-exposed infants, limited research explores the social and cultural barriers encountered by mothers living with HIV in rural health facilities in Ghana. This study investigates the challenges associated with exclusive breastfeeding and, where feasible, alternative replacement feeding among mothers living with HIV in rural areas, with a specific focus on how cultural beliefs influence adherence to recommended breastfeeding guidelines.

### Methods

This qualitative study was conducted in two rural community health facilities in Ghana, targeting mothers living with HIV who were actively seeking maternal and child health services. A purposive sampling technique was employed to ensure the inclusion of diverse experiences and perspectives, resulting in a sample of 32 respondents. Participants were selected based on their HIV status and engagement with maternal health services. Data collection involved in-depth interviews and inter-view guide as data collection tool designed to explore personal experiences and cultural influences on infant feeding practices. All interviews were transcribed verbatim and analysed using Colaizzi's descriptive phenomenological approach, facilitated by NVivo 12 software to ensure systematic data management and theme identification.

### Results

The study identified specific sociocultural barriers that hinder the prevention of mother-to-child transmission of HIV in rural communities. Key themes included

**Data availability statement:** The de-identified data set contains potentially sensitive patient information that could indirectly lead to the identification of individuals due to the nature of the health-related variables included. Access to the data is restricted to ensure the privacy and confidentiality of the individuals involved, in compliance with ethical guidelines for handling health data. These restrictions have been imposed by the Institutional Review Board (IRB) of the Committee on Human Research, Publication and Ethics, College of Health Sciences School of Medicine and Dentistry and the Ghana Health Service to safeguard participant confidentiality and uphold ethical research standards. For data access requests, please contact the following institutional body: Regional Health Directorate Ghana Health Service PMB, Bolgatanga, Ghana Phone: +233 0382024390 Email: uerrdhs@gmail.com

**Funding:** The author(s) received no specific funding for this work.

**Competing interests:** The authors have declared that no competing interests exist.

breastfeeding as a deeply rooted cultural practice, the customary use of water for infant feeding, traditional roles of mothers-in-law in infant care decisions, and food security concerns. Sub-themes further elaborated on breastfeeding as a long-standing tradition, perceived health benefits of breastfeeding, and respect for elders' advice, among others. These cultural norms and beliefs were found to significantly influence adherence to exclusive breastfeeding and, where feasible, alternative replacement feeding guidelines among mothers living with HIV.

## Conclusions

This study provides valuable insights into the sociocultural barriers affecting efforts to prevent mother-to-child transmission of HIV in Ghana rural areas. It highlights the need for culturally adaptive health policies and the development of community-based breastfeeding intervention programmes that align with traditional values while pro-moting safe feeding practices.

---

## Introduction

Preventing mother-to-child transmission (PMTCT) of HIV remains a critical global health challenge, particularly in sub-Saharan African countries where transmission occurs during pregnancy, childbirth, and breastfeeding [1,2]. Despite efforts, the United Nations Children's Fund [3] reported approximately 150,000 new child infections under 0–4 years old, falling short of the target of fewer than 20,000 new infections annually. This underscores the ongoing need for significant efforts to achieve the goals of PMTCT programmes.

The prevention of mother-to-child transmission programme in sub-Saharan Africa aims to reduce the transmission of HIV from mothers living with HIV to their infants during pregnancy, childbirth, and breastfeeding. This comprehensive approach includes antiretroviral therapy (ART) for pregnant women, safe infant feeding prac-tices, and counselling to support adherence and reduce stigma [4].

Breastfeeding is a primary mode of infant feeding in sub-Saharan Africa, but it poses challenges for mothers living with HIV due to the risk of HIV transmission through breast milk. Studies estimate that between 30% and 45% of perinatal HIV transmissions occur through breastfeeding [5]. In some communities, there is a strong preference for breastfeeding, and alternative feeding options may be stig-matised or considered unacceptable. This cultural norm can lead to mixed feeding practices, which increase the risk of HIV transmission [6].

Also, individual and household barriers affect exclusive breastfeeding among mothers living with HIV. Factors such as occupation, early motherhood, postpartum depression, and breast conditions like sores and abscesses can hinder exclusive breastfeeding. Household challenges, including lack of support from family members, and HIV-related stigma, may discourage mothers from adhering to recommended feeding practices [5,7].

Sub-Saharan Africa has made significant progress in PMTCT, with a substantial reduction in new HIV infections among children. Over the past 30 years, new HIV cases have dropped by 70% in the region [8]. This success is attributed to increased access to ART, improved healthcare infrastructure, and targeted interventions for pregnant women. However, challenges remain. The risk of HIV transmission through breastfeeding persists, especially when exclusive breastfeeding is not practiced as mixed feeding has been identified as factor influencing by individual circumstances, continues to be a significant barrier [9].

While PMTCT programmes in sub-Saharan Africa have achieved notable successes, addressing breastfeeding complexities and sociocultural barriers is crucial for further reducing perinatal HIV transmission. Tailored interventions that account for cultural contexts, provide comprehensive support to mothers living with HIV, and promote exclusive breastfeeding are essential to enhance the effectiveness of PMTCT efforts in the region [10,11].

Exclusive breastfeeding offers significant health benefits for infants born to mothers living with HIV, particularly in resource-limited settings. When combined with antiretroviral therapy (ART), exclusive breastfeeding significantly reduces the risk of mother-to-child HIV transmission. Research has demonstrated that with consistent maternal ART adherence, the risk of HIV transmission through breastfeeding can be reduced to less than 5% [12].

Beyond lowering the risk of HIV transmission, exclusive breastfeeding provides essential nutrients and antibodies that strengthen the infant's immune system, protecting against common childhood illnesses such as diarrhoea and pneumonia. This protection is particularly critical in areas with limited access to clean water and adequate sanitation, as it helps reduce the incidence of waterborne diseases [13].

In contrast, replacement feeding (using formula or animal milk instead of breast milk) completely eliminates the risk of HIV transmission through breastfeeding but introduces significant challenges. In resource-limited settings, replacement feeding is associated with increased risks of malnutrition, diarrhoea, and infant mortality due to unsafe water, poor sanitation, and limited healthcare access [14].

The World Health Organisation (WHO) recommends that mothers living with HIV breastfeed for at least 12 months and continue up to 24 months or longer while receiving comprehensive support to maintain ART adherence [15].

Therefore, while replacement feeding may be a viable alternative in contexts where it is safe, feasible, and sustainable, exclusive breastfeeding combined with ART adherence remains the recommended strategy in many parts of sub-Saharan Africa to optimise infant health outcomes and minimise the risk of HIV transmission.

According to WHO guidelines [16], mothers living with HIV are recommended to receive antiretroviral therapy (ARV) and exclusively breastfeed for six months postpartum, followed by complementary feeding unless safe conditions for replacement feeding are assured. These measures aim to reduce MTCT of HIV, with exclusively breastfeed significantly lowering transmission rates during the first six months of an infant's life. However, adherence barriers such as stigma and discrimination, lack of partner or family support, inadequate knowledge and counselling to ARV and exclusively breastfeed practices remains low in sub-Saharan African (SSA) countries, posing challenges to effective PMTCT [17,18]. Cultural perceptions of breastfeeding among pregnant and postpartum women complicate the adoption of exclusive breastfeeding and, where feasible, alternative replacement feeding in SSA [18,19].

Several studies have explored cultural practices related to pregnancy and childbirth, food beliefs, traditional harmful practices, and perinatal care among non-HIV-positive women [20–23]. However, these studies were conducted outside of the specific study area, highlighting a gap in the literature regarding cultural practices related to breastfeeding among HIV-positive pregnant and postpartum women in Northern Ghana.

This study investigates breastfeeding practices among pregnant and postpartum women living with HIV in rural communities in Northern Ghana, specifically examining adherence to exclusive breastfeeding and, where feasible, alternative replacement feeding during the first six months postpartum [24]. The study aims to inform policy interventions aimed at eliminating MTCT of HIV during the postpartum period, specifically within the first 12 months after childbirth.

## Research methods

### Study setting and design

The study was conducted in Bongo and Talensi District hospitals in the Upper East Region of Northern Ghana, located 14.8 km and 31 km from the regional capital, respectively. These districts face HIV/AIDS determinants such as high-risk sexual behaviour, low literacy rates, polygamy, widow inheritance, a nonchalant lifestyle, and migration. The study area was chosen for their rich cultural values and deprived rural status for the understanding of the phenomenon.

In this qualitative study, a phenomenological design was used to understand the phenomenon in its natural setting. Descriptive phenomenology is a qualitative research methodology aimed at capturing and describing the lived experiences of individuals regarding a specific phenomenon. Rooted in the philosophical traditions of Edmund Husserl, this approach focuses on understanding the essence of experiences from the first-person perspective, without the influence of the researcher's preconceptions or biases [25]. In this paper, descriptive phenomenology would involve collecting detailed narratives from mothers about their breastfeeding experiences, especially as they navigate the challenges posed by HIV prevention protocols.

A critical aspect of descriptive phenomenology is the practice of bracketing, epoché, and phenomenological reduction. Bracketing involves the researcher setting aside their own beliefs, judgments, and experiences to view the phenomenon purely from the participants' perspective [26]. This process ensures that the data collected were as close to the lived experience of the participants as possible, free from external interpretations. In this study, the researchers bracketed their own views on breastfeeding and HIV transmission to understand how the participants perceive and experience these issues.

Epoché, closely related to bracketing, is the process by which the researcher suspends all preconceived notions and biases, entering the research with a fresh perspective [25]. This state of openness allows the researcher to grasp the true essence of the participants' experiences. In this study, the researcher approached each interview without prior assumptions about cultural practices or healthcare barriers, allowing the mothers' authentic voices to emerge.

Phenomenological reduction, another core element of descriptive phenomenology, involves breaking down the experiences into their essential structures. It is a method of analysis that seeks to uncover the fundamental meaning of the experiences described by participants [27,28]. By focusing on the invariant aspects of these experiences, the researcher can identify the core themes and essences that define the phenomenon. In this paper, reduction would involve analysing the narratives of mothers living with HIV to distil the essential themes related to breastfeeding complexities and sociocultural barriers, providing a clearer understanding of their lived experiences. Data were collected from March 28, 2022, to June 28, 2022 through in-depth interviews with pregnant and postpartum women living with HIV using PMTCT services to explore sociocultural barriers to exclusive and, where feasible, alternative replacement feeding. The data were analysed using Colaizzi's [27] descriptive phenomenological technique.

### Participants and recruitment criteria

Participants in the study included pregnant and postpartum women living with HIV who had previously used the PMTCT service at the health facilities in these rural communities. The data collection cover three months, from March 28, 2022, to June 28, 2022, to identify the respondents for inclusion into the study. During this period, the registers were reviewed to select participants who met the inclusion criteria. The data collection for respondent inclusion began in March 2022 and continued until June 2022. This three-month period allowed the authors to gather the necessary data for study inclusion. Participants who were HIV-positive but not pregnant or breastfeeding were not allowed to participate in the study. The participants were contacted via phone, visited at their homes, and recruited on the day of their consultation. A purposive sampling method was employed to recruit 32 women from the rural community's health facilities. The sample size of 32 participants was determined using the concept of data saturation, which refers to the point at which no new information or themes are emerging from additional data collection. In qualitative research, data saturation is commonly used to decide

when enough participants have been included to sufficiently explore the research question [29]. The selection process was guided by the research objective, aiming to include women with diverse experiences related to the phenomenon under investigation. Health facility records and referrals from healthcare providers were used to identify potential participants who met the inclusion criteria. These women were then approached and informed about the study, and those who agreed to participate were included. The purposive sampling strategy ensured that participants with a wide range of experiences were selected to provide rich, in-depth insights into the phenomenon, thereby enhancing the quality and comprehensiveness of the findings [30].

## Data collection tool and procedure

The primary data collection tool for this study was an interview guide designed to explore cultural and social barriers to exclusive breastfeeding and, where feasible, alternative replacement feeding among mothers living with HIV in rural Ghana. The guide was developed using Colaizzi's descriptive phenomenology framework, which allowed for a thorough examination of participants' lived experiences and beliefs regarding infant feeding practices [27].

The interview guide was developed collaboratively, drawing on existing literature about breastfeeding practices, cultural influences on maternal health, and WHO [16] guidelines for infant feeding in the context of HIV. It consisted of open-ended questions aimed at encouraging participants to provide detailed responses. The flexible nature of the questions also allowed for follow-up questions during the interviews. The design of the guide prioritised culturally sensitive language and a non-judgmental approach to ensure participants felt comfortable sharing their experiences [31].

Key themes and questions in the interview guide included cultural practices and beliefs, such as: Can you describe the common breastfeeding practices in your community? and What beliefs do people have about breastfeeding and giving water to infants? The guide also explored infant feeding choices and influences with questions like, how did you decide how to feed your baby? and Who influences your feeding decisions the most (family members, health workers, elders)?

Also, the interview guide addressed the role of family and community, asking questions such as: What role do mothers-in-law or elders play in your breastfeeding decisions? It also touched on food security and resources, with questions like: Do you feel you have enough food to meet your and your baby's nutritional needs? and What challenges do you face in accessing nutritious food for yourself and your baby?"

Finally, the guide explored health beliefs and HIV-related practices, including: What concerns do you have about breastfeeding as a mother living with HIV? and how do you feel about the recommendations from healthcare workers regarding exclusive breastfeeding? These themes and questions were designed to comprehensively capture the social, cultural, and health-related factors that shape breastfeeding and replacement feeding practices among mothers living with HIV in rural Ghana.

The data were gathered through interviews, allowing for a flexible and conversational approach to understanding participants' experiences [32]. The data collection took place from March 28, 2022, to June 28, 2022, followed by participant follow-up and validation from July 22, 2022, to September 23, 2022. The interviews were conducted in an office assigned by the facility's unit head, but the researcher also made accommodations by meeting participants in their preferred locations, such as their homes, school grounds, and convenient hospital locations. This approach helped create a more comfortable and familiar environment for the participants, encouraging open and honest dialogue.

To address the sensitive nature of the study and reduce the risk of stigma, the researcher took great care to ensure privacy during interviews. Participants expressed a strong preference for privacy, and the researcher respected this by selecting discreet interview locations and scheduling sessions at times when participants felt most comfortable and secure [33]. This helped to foster a sense of trust and safety, which was crucial for collecting authentic and detailed information.

Before the interviews began, all participants who agreed to join the study were required to sign or thumbprint the consent form, ensuring they fully understood the purpose of the research and their rights as participants. The consent process included reading the information to those who needed assistance, with ample opportunity for participants to ask questions

and clarify any concerns [34]. This thorough consent process was essential in building trust and ensuring ethical standards were maintained throughout the study.

Anonymity was strictly upheld by assigning pseudonyms to all participants, ensuring that their identities remained confidential. The interviews typically lasted between 45 and 60 minutes, providing ample time for participants to express their experiences in detail [35]. The researcher employed a interview guide format, using open-ended questions to guide the conversation while allowing participants the freedom to discuss topics that were most relevant to them. This approach not only ensured the collection of rich, in-depth data but also allowed the researcher to explore emerging themes as they arose.

Interviews were conducted until data saturation was reached, meaning that the researcher continued to gather data until no new information or themes emerged [36]. This approach ensured that the data collection process was thorough and that the findings were representative of the full range of participants' experiences. Also, the researcher took detailed notes and used audio recording (with participants' consent) to capture the nuances of each interview, ensuring the accuracy and reliability of the data collected. This careful and systematic approach to data collection was integral to the study's overall rigor and credibility.

## Data analysis

In this study, a co-coder was involved in the data analysis process to enhance the reliability and accuracy of the findings [37]. The co-coder independently analysed a portion of the data, applying the same coding framework based on Colaizzi's descriptive phenomenology. Afterward, the primary coder and co-coder compared their codes and interpretations to ensure consistency and resolve any discrepancies. This collaborative approach helped to reduce bias, improve the validity of the identified themes and sub-themes, and ensure a more rigorous analysis of the qualitative data [38].

The data in this study was analysed using an inductive approach, where themes were derived directly from the participants' responses rather than from pre-existing theories [39]. Colaizzi's descriptive phenomenology was applied to identify patterns in the data. Through interviews, key themes, such as breastfeeding as a cultural practice and traditional infant water feeding, emerged naturally. The analysis focused on understanding the lived experiences of Mothers living with HIV, allowing themes and sub-themes to emerge from the data itself. NVivo 12 software was used to organize and facilitate this inductive analysis.

The Colaizzi [27] method, a popular data analysis technique in descriptive phenomenology, was utilized to analyse the data collected from participants. This method emphasizes a rigorous, structured approach to capturing the true essence of participants' lived experiences. The process began with familiarization, where the researcher read and re-read the interview transcripts multiple times to deeply understand the participants' perspectives and experiences.

Next, the researcher identified significant statements from the transcripts, selecting key phrases or sentences that directly related to the phenomenon being studied. These significant statements were extracted and organized in a word document template for easy reference. From these statements, meanings were carefully formulated, ensuring that the participants' intended meanings were accurately interpreted without imposing bias or misinterpretation.

The next step involved clustering formulated meanings into themes, where the researcher grouped similar meanings together to create broader, overarching themes that reflected common patterns across participants' experiences. These themes were then used to develop a detailed and exhaustive description of the phenomenon, ensuring that the depth and complexity of participants' experiences were fully represented.

Finally, the researcher conducted a validation process by contacting participants via personal visits or phone calls, ensuring that the findings accurately reflected their experiences. This validation step enhanced the credibility and reliability of the study's conclusions. The Colaizzi method provided a transparent and methodical approach to data analysis, ensuring that the research remained grounded in participants' authentic experiences. An example illustrating the development of theme clusters and emergent themes from significant statements has been provided in Table 1.

**Table 1. Development of clusters of themes and emergent themes from the significant statements and formulated meanings.**

| Significant Statements | Formulated Meanings | Theme Clusters | Emergent Themes |
|---|---|---|---|
| "Breastfeeding is something we have always done in our culture." | Breastfeeding is a long-standing practice passed down through generations. | Long-standing practices. Traditional health benefits of breastfeeding. | Breastfeeding as a cultural practice |
| We believe that giving infants water helps with digestion." | Providing infants with water is a traditional practice believed to aid in digestion | Water feeding improves infant's health. Ancestral customs and practices. | Traditional infants' water feeding as a cultural practice |
| "The elder women in the family are very influential in child-rearing practices." | Elder women, particularly mothers-in-law, have a strong influence on child-rearing practices. | Influence of elder women in the family. Respect for elders' advice. | The traditional and customary role of the mother-in-law |
| "There are times when we struggle to have enough food for everyone." | Food scarcity is a challenge that impacts family well-being. | Challenges in food availability. Psychological impact of food on medication. | Food security |

Source: Field Data transcribed (2022)

## Trustworthiness/rigour

To ensure trustworthiness in the qualitative research, the four components outlined by Lincoln and Guba and Lincoln [40] credibility, transferability, confirmability, and dependability were adequately addressed. Credibility was ensured through audio recording, verbatim transcription, and peer evaluation. Transferability was enhanced by providing detailed descriptions to facilitate application to similar work. Confirmability was maintained using bracketing to avoid personal bias. Dependability was ensured by using audit trails. Accuracy was enhanced through prolonged engagement, involving extended time with respondents in their native culture to understand their behaviour, values, and social relationships in context. Epoché improves reliability and validity in the study by ensuring that the researcher sets aside personal biases, assumptions, and preconceived ideas during data collection and analysis. This practice enhances reliability by promoting consistency in how data is interpreted and validity by ensuring the findings genuinely reflect the participants' perspectives rather than the researcher's viewpoints [25].

The principal investigator (PI) is originally from Northern Ghana, the setting of the study, and speaks the local language fluently. Although the PI shares cultural familiarity with the community, she currently resides and works outside the region and had no prior professional or personal relationships with the study participants or healthcare facilities involved.

This cultural and linguistic familiarity facilitated rapport-building and more open communication during interviews, contributing positively to the depth of data collected. However, to preserve objectivity and adhere to the principles of descriptive phenomenology, the PI practiced bracketing throughout the research process, actively setting aside personal biases and assumptions during data collection and analysis

## Ethical statement

This work was ethically approved by the Committee on Human Research, Publication, and Ethics, School of Medicine and Dentistry (CHRPE/AP/497/21), and the Regional Directorate of Health Services (RDHS/11/001). Permission to collect data from hospitals were also obtained and all participants provided written informed consent.

## Inclusivity in global research

Additional information regarding the ethical, cultural, and scientific considerations specific to inclusivity in global research is included in the Supporting Information (SX Checklist)

## Results

### Socio-demographic characteristics of the participants

### Presentation of results

According to the findings of the study, four major themes emerged from the data regarding breastfeeding complexities and sociocultural barriers in the context of preventing perinatal transmission of HIV. The emergent themes included breastfeeding as a cultural practice, "traditional infant's water feeding" as a cultural practice, and the customary role of the mother-in-law during breastfeeding, and lack of food. The Table 2 gives a summary of the emergent themes and cluster of themes of the study participants Table 3.

**Theme 1: Breastfeeding as a cultural practice.** Breastfeeding is a deeply rooted cultural practice, passed down through generations. It is highly valued for its health benefits, as it promotes infant health and strengthens the bond between mother and child.

***Breastfeeding as a long-standing practice*:** Mothers living with HIV shared their experiences of breastfeeding as a cultural obligation rather than a personal choice between exclusive breastfeeding or exclusive replacement feeding. According to traditional practices, a woman cannot choose whether or not to breastfeed, as it is culturally expected that she must breastfeed her baby. One of the participants said:

*Breastfeeding in prevention of mother-to-child transmission of HIV, the woman has the option to breastfeed or not breastfeed, but traditionally, thewoman has no choice, so you must breastfeed the baby...... If you wean thechild in a*

**Table 2. Socio-demographic characteristics.**

| Category | Details |
|---|---|
| Total number of Participants | 32 |
| Age range | Youngest: 20 years, Oldest: 47 years |
| Number of children per mother | Ranged from 0 to 4 children |
| HIV status of children | 14 children HIV positive, 8 children with unknown status (pending test results) |
| Exclusive breastfeeding practices | Only 5 mothers practiced exclusive breastfeeding despite wide-spread knowledge of benefits |
| Marital status | 22 married |
| | 5 single |
| | 3 widowed |
| | 2 divorced |
| Educational background | 1 with university education |
| | 1 with polytechnic education |
| | 7 with secondary education |
| | 7 with junior high education |
| | 4 with primary education |
| | 12 with no formal education |
| Religious affiliation | 26 Christian |
| | 6 Muslim |
| Occupations | 29 farmers |
| | 1 teacher |
| | 1 trader |
| | 1 unemployed |

**Table 3. The emergent and cluster themes.**

| Emergent themes | Cluster of themes |
|---|---|
| Breastfeeding as a cultural practice | • Breastfeeding as long-standing practices.<br>• Traditional health benefits of breastfeeding. |
| Traditional infants' water feeding as a cultural practice | • Beliefs it improves infant's health.<br>• Ancestral customs and practices |
| The traditional and customary role of the mother-in-law | • Influence of elder women in the family.<br>• Respect for elders' advice. |
| Food security | • Challenges in food availability.<br>• Psychological impact of food on medication |

Source: Field Data transcribed (2022)

*year, the mother-in-law will object, and other family members will insist that the child be breastfed for at least two years* (Nmabono, mother living with HIV)

***Traditional health benefits of breastfeeding:*** The traditional health benefits of breastfeeding include providing essential nutrients and antibodies that strengthen an infant's immune system and protect against illnesses. Breastfeeding also supports optimal growth and development, contributing to the baby's overall health and well-being. A participant said:

*Breastfeeding helps to keep the baby healthy and strong and you opt not to practice breastfeeding in the traditional family house.*"(Atipoka, mother living with HIV).

The traditional health benefits of breastfeeding include providing essential nutrients and antibodies that strengthen an infant's immune system and protect against illnesses. Breastfeeding also supports optimal growth and development, contributing to the baby's overall health and well-being.

*I feed the child whatever food the child can eat. I also give the child water. Many mothers do not do exclusive breast-feeding in this community here because of the farming activities during this season. They will be busy on their farms and will not always be available to breastfeed the child all the times. There is also the belief that you must breastfeed the child as a mother even if you can afford the formular food you cannot choose to do replacement feeding; if you choose to do so, your family members will not understand* (Talenmah, mother living with HIV).

**Theme 2: Traditional infants water feeding" as a cultural practice.** The term "traditional infants water feeding" refers to a cultural practice in northern Ghana where infants are given water forcefully, usually by their mothers or mothers-in-law, particularly in rural communities. This is illustrated in the two subthemes: Water feeding improves the infant's health and ancestral customs and practices.

***Water feeding improves infant's health:*** According to the participants this practice is often employed when a child is perceived to be unwell or unwilling to drink. Mothers-in-law believe that when a child cries, it indicates abdominal discomfort, and they prepare boiled herbs and leaves for the child to ingest, aiming to cleanse the intestines and strengthen the immune system.

*They bathed the child in water boiled with leaves and herbs after I gave birth. They also gave the child water, and the name is "nyuule" [to place the child on a woman's lapse and give water]. This was done to cleanse the child's intestines and strengthen the child. When I became pregnant and gave birth, I was fifteen years old* (Amiamah, mother living with HIV).

To the mothers, they felt that it is the mothers-in-law make it difficult to practice exclusive breastfeeding. They cannot also prevent them if they want to give water to the child, especially when the is sick. This was what she narrated:

*Mother-in-law occasionally "nyuules" the child [giving the child water using the traditional method] to make the child strong, and you cannot stop her from doing so. Family members can make it difficult to practise exclusive breastfeeding because they believe we were only giving water and did not die* (Apambila, mother living with HIV)

***Ancestral customs and practices:*** The practice of giving water to infants is rooted in ancestral customs and my mother-in-law always recommended their daughters-in-law to do so. This was what one of the participants said:

*The mothers-in-law do not always want their daughters-in-law tobreastfeed exclusively, but when educated, they sometimes understand but are unwilling to allow their daughters-in-law to practise it* (Yinemah, mother living with HIV).

**Theme 3: Traditional and customary role of mother-in-law.** In traditional societies, mothers-in-law play significant roles during their daughters-in-law's pregnancies and after childbirth. They provide guidance and influence decisions, particularly concerning feeding practices such as exclusive breastfeeding or replacement feeding. Their involvement significantly shapes maternal and child health practices. The following two subthemes illustrate this influence.

***Influence of elder's women in the family:*** According to the participants, elder women, particularly mothers-in-law, have a strong influence on child-rearing practices. One of the mothers said:

*My baby was fed with breastmilk and water. I did not breastfeed exclusively because my mother-in-law refused, claiming that giving the child barbal water makes the child healthy and strong* (Abonpoka, mother living with HIV)

The mothers-in-law do not want their daughters-in-law to practice exclusive breastfeeding because they feel like we are punishing the child. To demonstrate this, a participant narrated:

*Exclusive breastfeeding is not commonly practised in this community. Due to the mother-in-law's perception that we are punishing the child by denying him water, few mothers exclusively breastfeed their infants. The mother-in-law will argue that since you received water from me and you didn't die, why do you want to keep the child away from water?* (Lizzy, mother living with HIV)

The participants also reported that the role of mothers-in-law influences the positive mothers' practise of exclusive breast feeding. The mothers-in-law promote the use of herbs to improve the health of their children. This practise puts MTCT of HIV at risk. This was how Fatima said:

*The mothers-in-law are the issue because they insist on the child drinking water. When the mother gives birth, the mother-in-law sock herbs and givethem to the child to help strengthen the child's body* (Fatima, mother living with HIV)

***Respect for elders 'advice:*** One of the participants mentioned that during antenatal care, mothers were educated on the importance of exclusive breastfeeding, and some were aware of its benefits. However, many hesitate to practice it for fear of disobeying their mothers-in-law. Mothers living with HIV often prioritise their mothers-in-law's advice over guidance from health professionals. According to one of the participants:

*"My mother-in-law's advice is highly respected in our household, particularly in this community."* (Paulina, mother living with HIV)

**Theme 4: Food security.** Food security emerged as a critical issue affecting exclusive breastfeeding practices. Participants highlighted the essential role of adequate nutrition in sustaining breastfeeding. The two subthemes below give further illustration on this.

***Challenges in food availability:*** Many mothers reported insufficient food intake, which they believed directly affected their milk production. They expressed that better nutrition could improve milk supply, supporting exclusive breastfeeding. Concerns about inadequate milk production due to poor nutrition often led to supplementing breastfeeding with other foods and liquids. This was stated by one of the participants:

*There is not enough food for me to eat in order to produce enoughbreastmilk to feed thebaby. I am unable to practise exclusive breastfeedingbecause breastmilk is insufficient for the baby. So, I will have to feed the baby porridge and "zoomkoom* (Pogbila, mother living with HIV).

Participants stated that due to a lack of food, "*zoomkoom*" [local drink rich with vitamin C made from water] served as lactogogue for the mother and supplement drink for the child. When the baby is crying, the participants indicated that it is fed with "*zoomkoom*". It was also discovered that when lactating mothers use the "*zoomkoom*" as it stimulates the production of breastmilk to feed the baby. According to one participant:

*If you practise exclusive breastfeeding, you will not be able to give the baby "zoomkoom" again. There is insufficient food for you to eat in order toproduce enough breastmilk for the baby. We were told during antenatal careto practise exclusive breastfeeding, but there is not enough food for you to eat in order to produce enough breastmilk to breastfeed the baby* (Asaamah, mother living with HIV).

***Psychological impact of food on medication:*** One participant stated that food has become a major factor influencing exclusive breastfeeding and medication adherence. There is no food to eat when it is time for her to take the medications, and exclusive breastfeeding would be harmful to the baby. This was what she said:

*There is no food to eat before taking the medications, and you want me to exclusively breastfeed my child. Do you want my child to die? Mothers whopractise exclusive breastfeeding have a higher socioeconomic status* (Asumpoka, mother living with HIV)

A participant reported that a major challenge with exclusive breastfeeding in this community is the frequent complaints from mothers about insufficient food. Lactating mothers shared that while they often begin exclusive breastfeeding, they struggle to continue due to inadequate nutrition, which they believe affects their milk production. As a result, some mothers discontinue exclusive breastfeeding and switch to mixed feeding. The participant said:

*Exclusive breastfeeding is difficult for lactating mothers in this community. They frequently complained about a lack of food, which is the primary source of breastmilk. When asked, some are honest and say they don't practise exclusive breastfeeding due to a lack of food, while others pretend, they do. We do not have any food to give them to enforce it* (Ramatu, mother living with HIV).

## Discussions

The study explored exclusive breastfeeding complexities and sociocultural barriers in the prevention of perinatal transmission of HIV. It found that exclusive breastfeeding is hindered by practices such as infant water feeding, using zoomkoom as a lactagogue and food supplement. Also, cultural practices, the influence of mothers-in-law, and food security play significant roles in shaping breastfeeding practices among pregnant and postpartum women living with HIV.

## Breastfeeding as a cultural practice

The PMTCT guidelines recommend that mothers living with HIV exclusively breastfeed for the first six months before introducing complementary foods or replacement feeding. However, these mothers often view breastfeeding as a cultural obligation rather than a personal choice between exclusive breastfeeding and where feasible, alternative replacement feeding. Breastfeeding decisions are strongly influenced by family dynamics and cultural practices, with mothers-in-law playing a central role in decision-making.

The findings reveal that postpartum women living with HIV consider breastfeeding a cultural practice shaped by factors such as infant water feeding, the use of zoomkoom as a lactogogue and food supplement, and cultural beliefs also suggest that breast milk makes infants thirsty.

Mothers living with HIV reported feeling obligated to breastfeed because formula feeding or exclusive replacement feeding is not considered an option in their community. Mothers living with HIV do not use formula feeding, which increases the risk of infant HIV infections due to unsafe environmental and social conditions. These findings align with reports that postnatal HIV transmission during breastfeeding is a major concern in low- and middle-income countries, particularly in Sub-Saharan Africa, where breastfeeding is the only feasible, safe, and culturally acceptable option [1,41]. The findings also support the WHO [42] report that HIV transmission from mother to child typically occurs during pregnancy, childbirth, and breastfeeding.

The study's findings contrast with the WHO [15] report, which attributes the successful elimination of mother-to-child HIV transmission in countries like Sri Lanka and Thailand to medication adherence and formula feeding. This divergence may result from cultural norms in Sub-Saharan Africa that prioritize breastfeeding as the only socially acceptable feeding method, compounded by widespread poverty. Education also influences these regional differences. The study highlights inadequate adherence to infant feeding guidelines, with cultural barriers such as the use of zoomkoom as a lactogogue, beliefs about foremilk safety, and perceptions that breast milk induces infant thirst hindering efforts to prevent HIV transmission. Supporting the theory of planned behaviour, the study emphasizes the need for education and family involvement in PMTCT programmes to encourage alternative feeding methods. It also recommends expanding PMTCT efforts beyond hospitals to include community outreach, which could improve outcomes.

## Traditional infants' water feeding as a cultural practice

The study highlights how the cultural practice of "traditional infant water feeding" influences exclusive breastfeeding among mothers living with HIV. Exclusive breastfeeding is vital for PMTCT efforts, aligning with WHO and UNAIDS goals to eliminate mother-to-child HIV transmission and reduce HIV-related child morbidity. This practice involves offering water to newborns, based on the belief that infants are born thirsty and need water from birth. Mothers-in-law often promote this practice, complicating efforts to maintain exclusive breastfeeding.

The study reveals that cultural beliefs and family influences present significant barriers to exclusive breastfeeding among mothers living with HIV, despite its importance for PMTCT efforts. Participants disclosed that when a child is sick or reluctant to drink water, mothers or mothers-in-law often use herbal water, compelling intake by placing the child on their lap. Similarly, persistent infant crying is frequently attributed to abdominal pain, leading mothers-in-law to promote the use of local herbs for intestinal cleansing and immune support. These entrenched practices undermine exclusive breastfeeding for six months, despite its known benefits and educational efforts promoting it. The findings emphasize the need for culturally sensitive approaches and community engagement to address these challenges and support optimal infant feeding practices.

The findings align with those of Fauk et al. [43], who observed that cultural practices influence behaviours affecting HIV transmission, including within spousal relationships. Similarly, the World Health Organization [44] acknowledges the significant impact of cultural practices in African contexts, highlighting their implications for preventing mother-to-child

HIV transmission. Rodriguez et al. [45] also identified cultural practices, HIV-related stigma, and disclosure challenges as barriers to effective PMTCT programmes. Likewise, Malindi, Maputle, and Nemathaga [46] emphasised how traditional, cultural, and religious practices, along with mixed feeding, heighten the risk of HIV transmission to infants. Their findings stress the need for targeted education to address harmful cultural norms, particularly in rural communities where suboptimal exclusive breastfeeding practices hinder progress toward UNAIDS 90-90-90 targets for children.

## Traditional and customary role of mothers-in-law

Mothers-in-law play a significant role in reproductive and childcare practices in traditional African societies due to their status as experienced elder women responsible for family health and decision-making [47]. Their influence extends to guiding daughters during pregnancy and childbirth, often determining whether exclusive breastfeeding or exclusive replacement feeding is practiced. These findings align with Falnes et al. [47], who also highlighted the central role of mothers-in-law in health-related family decisions.

The study found that mothers-in-law often encourage early supplementation with other foods and water couple with regional factors, such as sunny weather conditions, further contribute to giving water to infants. These findings align with research by Kubuga and Tindana [48], which reported low exclusive breastfeeding rates in the Upper East Region, where the study was conducted. Despite WHO (2016) guidelines promoting exclusive breastfeeding for HIV-exposed infants, the practice remains limited among HIV-positive pregnant and postpartum women in the area.

The study aligns with recent reports byKyei-Arthur et al. [49] and Kubuga and Tindana [48], indicating a decline in exclusive breastfeeding practices. In response, the Ghana Health Service and Ministry of Health, in collaboration with WHO and UNICEF, are promoting awareness of its importance. The study reinforces WHO's position that breastmilk is the optimal infant food due to its safety, hygiene, antibodies that protect against childhood illnesses, and nutritional completeness for the first two years of life. However, many infants still do not receive exclusive breastfeeding for the recommended six months.

Additionally, the findings highlight those traditional expectations, especially from mothers-in-law, often conflict with recommended feeding practices for mothers living with HIV, such as exclusive breastfeeding or replacement feeding [47]. This contrasts with UNAIDS [50], which suggests that male family members may also influence maternal feeding decisions, underscoring the diverse factors affecting adherence to infant feeding guidelines.

The study identifies cultural attitudes and social norms as key factors influencing exclusive breastfeeding among mothers living with HIV. Practices like the use of herbs and liquid foods for infants often discourage exclusive breastfeeding. These findings align with previous studies by Laksono et al. [51] and Gabriel et al. [52], which link practices like the use of herbs and liquid foods for infants often discourage exclusive breastfeeding.

The study emphasizes the importance of exclusive breastfeeding in low-income communities for mothers living with HIV, as it reduces the risk of mother-to-child transmission (MTCT) of HIV through breast milk. Early weaning at six months is recommended to further lower this risk, but cultural norms favouring extended breastfeeding can contradict this approach and heighten the risk of MTCT. Consistent with WHO [42] findings, the study highlights how cultural norms can impact exclusive breastfeeding, underscoring the need for culturally sensitive PMTCT interventions tailored to local practices to minimize HIV transmission risks.

## Food security

The study identified food security as a crucial factor affecting exclusive breastfeeding practices among participants. Many mothers reported food insecurities that impacted their ability to produce sufficient breast milk, making exclusive breastfeeding difficult. They believed that better food access would improve milk production, enabling them to exclusively breastfeed their infants. However, actual or perceived food shortages often led mothers to supplement breast milk with

other foods and water. These findings emphasize the need for comprehensive support systems that address food security alongside breastfeeding promotion to improve maternal and child health outcomes.

The study confirmed that mothers living with HIV often reported insufficient breast milk production, with inadequate nutrition cited as a primary barrier to optimal breastfeeding [19]. However, it contradicted previous literature that also attributed breastfeeding challenges to factors such as lack of community support and breast infections related to poor breastfeeding practices [18,19].

Participants described using zoomkoom, a millet-based drink rich in vitamin C, as a substitute for breast milk when food was scarce. They believed that feeding zoomkoom to their babies served as food supplement and stimulated breast milk production among lactating mothers. These findings align with previous research indicating that women may give birth at home to access culturally accepted foods like zoomkoom [53]. The participants highlighted that food availability plays a significant role in their ability to practice exclusive breastfeeding and adhere to HIV medications. This similarity suggests continuity in findings within the same study context.

The study identified food insecurity as a significant barrier to exclusive breastfeeding in the community, a concern raised by both healthcare providers and lactating mothers. Many mothers reported starting exclusive breastfeeding but struggled to continue due to insufficient food, which hindered breast milk production and led some to switch to mixed feeding. These findings align with previous research linking food security to breastfeeding practices among mothers living with HIV [54,55]. However, the study did not support other factors, such as poor policy dissemination, knowledge gaps, misunderstandings about exclusive breastfeeding, lack of counselling, or stigma related to HIV transmission fears, which have been identified as barriers in some studies [54,56–59].

## Conclusions

The study found that exclusive breastfeeding practices among pregnant and postpartum women living with HIV in the Talensi and Bongo District Hospitals were significantly influenced by cultural practices, traditional infant feeding methods, the customary role of mothers-in-law, and food security.

The study provides a comprehensive understanding of contextual barriers, offering a detailed understanding of the multifaceted challenges mothers face in adhering to recommended breastfeeding practices. By highlighting cultural practices like traditional infant feeding methods and the influential role of mothers-in-law, the study provides valuable insights into deeply rooted beliefs and behaviours shaping breastfeeding decisions within these communities. This cultural context enriches the study by revealing factors that may not be evident in other settings.

Moreover, the study's findings are highly relevant to global health goals, particularly those focused on eliminating mother-to-child HIV transmission through breastfeeding interventions. By aligning with these objectives, the study contributes to broader efforts in global health strategies, underscoring the importance of culturally sensitive approaches for achieving positive maternal and child health outcomes. The application of the descriptive phenomenological research design enhances the study's methodology offering a thematic analysis of health beliefs and behaviours related to breastfeeding. This theoretical approach not only deepens the understanding of factors influencing breastfeeding practices but also enhances the study's utility in informing public health interventions and policies.

Also, the study proposes practical policy recommendations for inclusive PMTCT programmes and community-based interventions. These recommendations, rooted in the study's findings, aim to effectively address local contexts and barriers. This approach holds promise for potentially improving breastfeeding practices among Mothers living with HIV in similar rural settings.

## Limitations of the study

The study provides valuable insights but also has several limitations. First, its findings may not be widely applicable due to the specific cultural and social contexts of the rural study area, which limits their generalisability to other regions or

populations with different sociocultural dynamics. The focus on rural communities and specific healthcare facilities also introduces potential sampling bias, possibly overlooking diverse perspectives beyond these settings.

Also, the evolving nature of health policies and cultural practices means the study's findings may have limited temporal relevance as policies and societal norms change. Also, the qualitative insights are valuable, the lack of quantitative data on the prevalence and impact of cultural beliefs and practices related to breastfeeding limits the study's ability to assess broader population trends and quantitative outcomes. Lastly, while the researcher maintained reflexivity and adhered to bracketing techniques throughout the research process, complete detachment may not be fully achievable.

## Supporting information

**S1 File. Inclusivity-in-global-research-questionnaire.**
(DOCX)

## Acknowledgments

To all staff of the Department of Sociology and Social work, College of Health Sciences, School of Medicine and Dentistry Committee on Human Research, Publication and Ethics, Kwmane Nkrumah University of Science & Technology, Regional Health Directorate of Ghana Health Services, Upper East Region, and to all the clients and the staff of the Bolgatanga Regional Hospital, Bongo District Hospital and Talensi District Hospital without you this thesis would not have been possible.

## Author contributions

**Conceptualization:** Awinaba Amoah Adongo.

**Methodology:** Kofi Osei Akuoko, Jonathan Mensah Dapaah, Esmeranda Manful.

**Supervision:** Kofi Osei Akuoko, Jonathan Mensah Dapaah, Esmeranda Manful.

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
