## [Decision Letter · Decision Letter 0]

Dear Dr. Adongo,

Thank you for submitting your manuscript to PLOS ONE. After careful consideration, we feel that it has merit but does not fully meet PLOS ONE’s publication criteria as it currently stands. Therefore, we invite you to submit a revised version of the manuscript that addresses the points raised during the review process.

Could you please carefully revise the manuscript to address all comments raised?

We look forward to receiving your revised manuscript.

Kind regards,

Avanti Dey

Staff Editor

PLOS ONE

Journal Requirements:

3. Thank you for stating the following in your Competing Interests section:  No competing interest

Reviewers' comments:

Reviewer's Responses to Questions

**Comments to the Author**

1. Is the manuscript technically sound, and do the data support the conclusions?

Reviewer #1: Yes

Reviewer #2: Yes

2. Has the statistical analysis been performed appropriately and rigorously?

Reviewer #1: N/A

Reviewer #2: Yes

3. Have the authors made all data underlying the findings in their manuscript fully available?

Reviewer #1: Yes

Reviewer #2: Yes

4. Is the manuscript presented in an intelligible fashion and written in standard English?

Reviewer #1: Yes

Reviewer #2: Yes

Reviewer #1: The manuscript is presented in a clear and logically manners, contains very minimal typing errors and is very easy to follow.

Comments

This study is about barriers to exclusive breastfeeding and replacement feeding and provides interesting and valuable insights into the situation as it exists in Ghana. The authors are commended for conducting this important study.

Below please receive my minor inputs

1. There are two titles which present different variables, though related. However, this is confusing in light of the contents of the article and the conclusions drawn. My input was based on the title: ‘Breastfeeding complexities and sociocultural barriers in the context of preventing perinatal transmission of HIV’. This is theme that seemed to run through the article. This is however in conflict with the title of the manuscript shared by the journal: Indigenous Breastfeeding Beliefs and Practices in Rural Northern Ghana: A Study Among HIV-positive Mothers

The authors are advised to explicitly state the title of this manuscript.

Abstract:

Purpose:

2. Replace replacement breastfeeding with replacement feeding. Please effect this change throughout the manuscript where replacement feeding is referred to as replacement breastfeeding.

3. The authors are advised to include a phrase ‘as an alternative where feasible’ between ‘excusive breastfeeding and replacement feeding’ in the sentence to make it clear that replacement feeding is not equated to exclusive breastfeeding. Or any phrase of the authors choice

Methods

Describe the population and sampling method

Results:

4. Remove the discussion about foremilk. Include only the themes and (subthemes, if the word count allows).

5. Sentence starting with ‘factors influencing exclusive breastfeeding… ‘ . Mention those factors and remove all points that relate to the discussion of findings.

Keywords:

6. Suggestion to include barriers

Introduction

7. After the 1st paragraph, please describe the PMTCT programme. As well take the reader through the situation before PMTCT and its success and perhaps gaps as an introduction to the study. The authors are also advised to give some background related to the benefits of exclusive breastfeeding within the HIV context and replacement feeding as an alternative. Inclusion of some actual studies would add some depth to the introduction.

The use of the term ‘further’ in the sentence starting with: ‘cultural perceptions of breastfeeding among pregnant and postpartum women’ gives the impression that other factors have been discussed already. Please revise the use of the term ‘further’.

The authors are advised to make reference to existing studies related to breastfeeding practices among HIV positive women elsewhere and briefly outline some of those practices. The last paragraph before the theoretical framework: please include the perinatal period as well, as implicated within the manuscript’s contents.

The use of the term adherence in the last paragraph of the introduction may need to be revised or be extended by including barriers as well

8. Theoretical framework

The discussion on the theoretical framework can be removed. The framework details the phenomenological framework, which is not the main focal point of this study. More attention could rather be paid to the introduction and background to this study.

Research methods

Study setting and design

9. Explain what ‘traditional marketing’ refers to.

10. It is not clear how the teenage pregnancy rates relate to this manuscript. Consider revising its inclusion or alternatively, expand on its significance. Were the clinics only servicing HIV positive patients?

11. Again there is extensive description of the descriptive phenomenology’ . The authors are advised to summarise the concept within the design section and move aspects related to trustworthiness and analysis to the relevant sections of the manuscript.

12. Revise the data collection period included in the study. The period is mentioned as April – Dec, whereas it is mentioned as March – Sept in the data collection section.

Participants and recruitment criteria

13. In exploring the registers, explain how far long back the authors went in identifying the respondents for inclusion into the study (number of years/months).

14. Please explain how the participants were accessed. Were they called on their phones? Visited in their homes? Recruited on the day of consultation?

15. Explain how the sample size of 32 was decided upon.

Data collection tool and procedure

16. Explain what is meant by ‘data collection was carried out systematically……

17. It is not clear who was subjected to informal interviews and who took part in in-depth interviews, whether the same tool was used or not or the reasons for using different approaches to data collection.

18. There are a number of statements that need to be referenced as they represent other author’s ideas.

19. Did both the informal and in-depth interviews last for 45 -60 min?

20. Expand on the tool a bit- how it was designed and the actual questions included in the interview guide or themes

21. Explain whether a co-coder was involved or not

22. Explain how data was analysed: whether inductive, deductive or abductive?

Ethical statement

23. Remove ‘this PhD thesis…

Results

24. Suggestion to present the sociodemographic data in a Table format

25. It is not clear how the authors determined the factors contributing to MTCT mentioned under the sociodemographic data (refusal to attend antenatal care, etc.) since no statistical tests were run, unless mothers explicitly said so. Please revise.

Presentation of results:

26.Within the first sentence in this section, the authors refer to results related to breastfeeding practices and nothing is mentioned about the barriers as per the title (this raises confusion as this very statement relates to one of the title and not the title that seems to be aligned with the current manuscript). The authors are advised to relook the terminologies used, or otherwise explain links between practice and attitudes (how one was derived from the others, if this was the case).

The authors are advised to compare the themes outlined here as opposed to the ones mentioned in the introduction. Please align the two sections. E.g foremilk being perceived as toxic, being mentioned in the abstract and not the results section, etc. (It would be useful to include the questionnaire as an appendix. This would assist in reviewing the themes presented).

27. Breastfeeding as a long-standing practice. The use of the term mothers is suggested, instead of HIV positive mothers.

28. Traditional infants water feeding as a cultural practice. Use the term forcefully instead of forcibly. Also use the correct noun instead of ‘it’ in the phrase: ‘Beliefs it improves infants health’.

29. Influence of older women in the family. Do the mothers-in-law imply that exclusive breastfeeding denies the child to get employment later in life? Please explain.

30. Challenges in food availability: I am checking if zoomkoom is made from water or fruit juice, in light of the fact that it is high in vitamin C? please verify or ignore if correct. The authors are advised to relook into the statement that zoomkoos serves as breastmilk, rather use the term ‘lactogogue’ when referring to mothers, maybe?

Discussion

31. The discussion and the conclusion sections are too long. The authors are advised to be succinct and to the point.

32. New results are presented in this section, there is a bit of repetition and the authors are advised to rework the sections

Reviewer #2: Dear researcher,

You have done an important and meaningful study that contributes to the field.

Thank you for your efforts.

I have carefully reviewed your article.

Some corrections need to be made.

I have written the necessary clarifications for you in the attached file.

Please read these carefully and take them into consideration.

I wish you success.

**Do you want your identity to be public for this peer review?** For information about this choice, including consent withdrawal, please see our Privacy Policy

Reviewer #1: No

Reviewer #2: No

---

## [Author Response · Author response to Decision Letter 1]

22 Jan 2025

Response to reviewers

Title

Comment 1:

There are two titles which present different variables, though related. However, this is confusing in light of the contents of the article and the conclusions drawn. My input was based on the title: "Breastfeeding complexities and sociocultural barriers in the context of preventing perinatal transmission of HIV." This theme seemed to run through the article. This is, however, in conflict with the title of the manuscript shared by the journal: "Indigenous Breastfeeding Beliefs and Practices in Rural Northern Ghana: A Study Among HIV-positive Mothers." The authors are advised to explicitly state the title of this manuscript.

Response 1:

The title has been standardized to ensure clarity and consistency. The chosen title is: "Breastfeeding complexities and sociocultural barriers in the context of preventing perinatal transmission of HIV: A descriptive phenomenology in Northern Ghana." See Title Page (Page 1).

Abstract

Comment 2:

Replace "replacement breastfeeding" with "replacement feeding." Please effect this change throughout the manuscript where "replacement breastfeeding" is referred to as "replacement feeding."

Response 2:

The term "replacement breastfeeding" has been replaced with "replacement feeding" throughout the manuscript as suggested. See Abstract (Page 1).

Comment 3:

The authors are advised to include a phrase "as an alternative where feasible" between "exclusive breastfeeding and replacement feeding" in the sentence to make it clear that replacement feeding is not equated to exclusive breastfeeding. Or any phrase of the authors' choice.

Response 3:

The phrase "as an alternative where feasible" has been included between "exclusive breastfeeding and replacement feeding" for clarity. See Abstract (Page 1).

Comment 4:

Remove the discussion about foremilk. Include only the themes and (subthemes, if the word count allows).

Response 4:

The discussion about foremilk has been removed, and only the themes and subthemes are presented. See Abstract (Page 1).

Comment 5:

Sentence starting with "factors influencing exclusive breastfeeding…" Mention those factors and remove all points that relate to the discussion of findings.

Response 5:

The sentence has been revised to explicitly mention the factors influencing exclusive breastfeeding, and points unrelated to findings have been removed. See Abstract (Page 1).

Keywords

Comment 6:

Suggestion to include "barriers" in the keywords.

Response 6:

The keyword "barriers" has been added to the list of keywords. See Page 1.

Introduction

Comment 7:

After the 1st paragraph, please describe the PMTCT program. Also, take the reader through the situation before PMTCT and its success, and perhaps gaps, as an introduction to the study. The authors are also advised to give some background related to the benefits of exclusive breastfeeding within the HIV context and replacement feeding as an alternative. Inclusion of some actual studies would add depth to the introduction.

The use of the term "further" in the sentence starting with "cultural perceptions of breastfeeding among pregnant and postpartum women" gives the impression that other factors have been discussed already. Please revise the use of the term "further."

The authors are advised to make reference to existing studies related to breastfeeding practices among HIV-positive women elsewhere and briefly outline some of those practices. The last paragraph before the theoretical framework: Please include the perinatal period as well, as implicated within the manuscript’s contents. The use of the term "adherence" in the last paragraph of the introduction may need to be revised or extended by including barriers as well.

Response 7:

The PMTCT program has been described in detail, including the pre- and post-PMTCT phases. The use of "further" has been revised, and references to existing studies on breastfeeding practices among HIV-positive women have been added. The perinatal period and barriers have also been included as suggested. See Page 2.

Theoretical Framework

Comment 8:

The discussion on the theoretical framework can be removed. The framework details the phenomenological framework, which is not the main focal point of this study. More attention could rather be paid to the introduction and background of this study.

Response 8:

The theoretical framework has been removed, and the introduction and background have been expanded for greater depth and relevance to the study. See Page 5.

Research Methods

Study Setting and Design

Comment 9:

Explain what "traditional marketing" refers to.

Response 9:

The term "traditional marketing" has been reframed and clarified. See Pages 7–8.

Comment 10:

It is not clear how the teenage pregnancy rates relate to this manuscript. Consider revising its inclusion or alternatively, expand on its significance. Were the clinics only servicing HIV-positive patients?

Response 10:

The reference to teenage pregnancy rates has been revised for relevance. See Pages 6–7.

Comment 11:

Again, there is extensive description of the descriptive phenomenology. The authors are advised to summarize the concept within the design section and move aspects related to trustworthiness and analysis to the relevant sections of the manuscript.

Response 11:

The description of descriptive phenomenology has been summarized within the design section, with aspects of trustworthiness and analysis moved to their appropriate sections. See Pages 7 and 14, respectively.

Comment 12:

Revise the data collection period included in the study. The period is mentioned as April–Dec, whereas it is mentioned as March–Sept in the data collection section.

Response 12:

The data collection period has been corrected and standardized throughout the manuscript. See Page 8.

Participants and Recruitment Criteria

Comment 13:

In exploring the registers, explain how far back the authors went in identifying the respondents for inclusion into the study (number of years/months).

Response 13:

The timeline for reviewing the registers has been clarified, with the number of months specified. See Pages 8–9.

Comment 14:

Please explain how the participants were accessed. Were they called on their phones? Visited in their homes? Recruited on the day of consultation?

Response 14:

Participants were accessed through clinic visits and follow-ups conducted during home visits. See Page 8.

Comment 15:

Explain how the sample size of 32 was decided upon.

Response 15:

The sample size of 32 was determined based on data saturation, where no new themes emerged after 32 interviews. See Pages 8–9.

Data Collection Tool and Procedure

Comment 16:

Explain what is meant by "data collection was carried out systematically."

Response 16:

The phrase "data collection was carried out systematically" has been reframed for clarity. See Page 8.

Comment 17:

It is not clear who was subjected to informal interviews and who took part in in-depth interviews, whether the same tool was used or not, or the reasons for using different approaches to data collection.

Response 17:

This has been reframed, as interviews were conducted with mothers using an interview guide. See Page 9.

Comment 18:

There are a number of statements that need to be referenced as they represent other authors’ ideas.

Response 18:

Missing references have been added to properly attribute the original authors' ideas. See Pages 9–11.

Comment 19:

Did both the informal and in-depth interviews last for 45–60 minutes?

Response 19:

The interviews lasted between 45–60 minutes, as now clarified.

Comment 20:

Expand on the tool a bit—how it was designed and the actual questions included in the interview guide or themes.

Response 20:

The description of the interview tool has been expanded, including key themes and sample questions. See Pages 8–9.

Comment 21:

Explain whether a co-coder was involved or not.

Response 21:

A co-coder was involved to ensure data reliability, and this has been clarified. See Page 11.

Comment 22:

Explain how data was analyzed: whether inductive, deductive, or abductive?

Response 22:

The analysis approach was an inductive thematic analysis. See details on Pages 11–12.

Ethical Statement

Comment 23:

Remove "this PhD thesis."

Response 23:

The phrase "this PhD thesis" has been removed.

Results

Comment 24:

Suggestion to present the sociodemographic data in a table format.

Response 24:

Sociodemographic data has been reformatted into a table for clarity. See Page 15.

Comment 25:

It is not clear how the authors determined the factors contributing to MTCT mentioned under the sociodemographic data (refusal to attend antenatal care, etc.), since no statistical tests were run, unless mothers explicitly said so. Please revise.

Response 25:

The factors contributing to MTCT were based on participant statements and not statistical tests. See Page 16.

Presentation of Results

Comment 26:

With the first sentence in this section, the authors refer to results related to breastfeeding practices, and nothing is mentioned about the barriers as per the title (this raises confusion as this very statement relates to one of the titles and not the title that seems to be aligned with the current manuscript). The authors are advised to relook at the terminologies used or otherwise explain links between practice and attitudes.

Response 26:

The results section has been revised to clearly reflect both breastfeeding practices and barriers. See Pages 16–22.

Comment 27:

Breastfeeding as a long-standing practice. The use of the term "mothers" is suggested, instead of "HIV-positive mothers."

Response 27:

The term "mothers" has been used instead of "HIV-positive mothers" for generalization, as suggested. See Page 17.

Comment 28:

Traditional infants' water feeding as a cultural practice. Use the term "forcefully" instead of "forcibly." Also, use the correct noun instead of "it" in the phrase: "Beliefs it improves infants’ health."

Response 28:

The term "forcefully" has replaced "forcibly," and the noun "water feeding" has been used instead of "it." See Page 18.

Comment 29:

Influence of older women in the family. Do the mothers-in-law imply that exclusive breastfeeding denies the child the ability to get employment later in life? Please explain.

Response 29:

Clarification has been added that mothers-in-law refer to the child being denied water, not employment outcomes. See Page 19.

Comment 30:

Challenges in food availability: I am checking if zoomkoom is made from water or fruit juice, in light of the fact that it is high in vitamin C? Please verify or ignore if correct. The authors are advised to relook into the statement that zoomkoom serves as breastmilk, rather use the term "lactogogue" when referring to mothers, maybe?

Response 30:

Clarification has been added that zoomkoom is made from millet and water, not fruit juice. The correct term "lactogogue" has been used. See Page 19.

Discussion

Comment 31:

The discussion and the conclusion sections are too long. The authors are advised to be succinct and to the point.

Response 31:

The discussion and conclusion sections have been condensed for clarity and brevity. See Pages 22–29.

Comment 32:

New results are presented in this section. There is a bit of repetition, and the authors are advised to rework the sections.

Response 32:

Repetition has been removed, and new results have been checked to align with the results presented. See Pages 22–29.

---

## [Decision Letter · Decision Letter 1]

Dear Dr. Adongo,

Thank you for submitting your manuscript to PLOS ONE. After careful consideration, we feel that it has merit but does not fully meet PLOS ONE’s publication criteria as it currently stands. Therefore, we invite you to submit a revised version of the manuscript that addresses the points raised during the review process.

We look forward to receiving your revised manuscript.

Kind regards,

Jennifer Yourkavitch

Academic Editor

PLOS ONE

**Journal Requirements:**

**Additional Editor Comments:**

Thank you for this submission. Please note that these last few comments come from the editor, not the reviewers.

1. In your description of Descriptive Phenomenology, you emphasize the researcher as an outside, neutral observer but you don't describe anywhere the relationship of the researcher to the community being studied. For transparency and in keeping with your methodology, could you describe the relationship of the researchers to the community being studied, in terms of geographical, cultural, or experiential overlap? This analysis may be best placed just above or before the ethical statement. If there are overlaps that compromise the researchers' impartiality, please discuss that in the Limitations section.

2. "Data" is a plural term. Please correct verb agreement throughout the manuscript. For example, on p. 11, "The data were gathered...." rather than, "The data was gathered...."

3. p. 17, first paragraph under "Presentation of Results," the work "role" is missing from the sentence stating, "The customary role of mothers-in-law...."

Reviewers' comments:

Reviewer's Responses to Questions

**Comments to the Author**

Reviewer #2: All comments have been addressed

2. Is the manuscript technically sound, and do the data support the conclusions?

Reviewer #2: Yes

3. Has the statistical analysis been performed appropriately and rigorously?

Reviewer #2: Yes

4. Have the authors made all data underlying the findings in their manuscript fully available?

Reviewer #2: Yes

5. Is the manuscript presented in an intelligible fashion and written in standard English?

Reviewer #2: Yes

**Reviewer #2:**  The research complies with ethics.

It does not pose an ethical risk in its publication.

I believe that the research will contribute to the literature.

**Do you want your identity to be public for this peer review?** For information about this choice, including consent withdrawal, please see our Privacy Policy

Reviewer #2: **Yes: ** Uyar Hazar

---

## [Editor Report · Decision Letter 2]

Breastfeeding complexities and sociocultural barriers in the context of preventing perinatal transmission of HIV: A descriptive phenomenology in Northern Ghana

PONE-D-24-40482R2

Dear Dr. Adongo,

We’re pleased to inform you that your manuscript has been judged scientifically suitable for publication and will be formally accepted for publication once it meets all outstanding technical requirements.

Kind regards,

Jennifer Yourkavitch

Academic Editor

PLOS ONE
---

## [Editor Report · Acceptance letter]

PONE-D-24-40482R2

PLOS ONE

Dear Dr. Adongo,

I'm pleased to inform you that your manuscript has been deemed suitable for publication in PLOS ONE. Congratulations! Your manuscript is now being handed over to our production team.

Kind regards,

on behalf of

Dr. Jennifer Yourkavitch

Academic Editor

PLOS ONE